

# Effects of temperature on metabolic scaling in black carp

Qian Li, Xiaoling Zhu, Wei Xiong, Yanqiu Zhu, Jianghui Zhang, Pathe Karim Djiba, Xiao Lv and Yiping Luo

Key Laboratory of Freshwater Fish Reproduction and Development, Ministry of Education, School of Life Sciences, Southwest University, Chongqing, China

## ABSTRACT

The surface area (SA) of organs and cells may vary with temperature, which changes the SA exchange limitation on metabolic flows as well as the influence of temperature on metabolic scaling. The effect of SA change can intensify (when the effect is the same as that of temperature) or compensate for (when the effect is the opposite of that of temperature) the negative effects of temperature on metabolic scaling, which can result in multiple patterns of metabolic scaling with temperature among species. The present study aimed to examine whether metabolic scaling in black carp changes with temperature and to identify the link between metabolic scaling and SA at the organ and cellular levels at different temperatures. The resting metabolic rate (RMR), gill surface area (GSA) and red blood cell (RBC) size of black carp with different body masses were measured at 10 °C and 25 °C, and the scaling exponents of these parameters were compared. The results showed that both body mass and temperature independently affected the RMR, GSA and RBC size of black carp. A consistent scaling exponent of RMR (0.764, 95% CI [0.718–0.809]) was obtained for both temperatures. The RMR at 25 °C was 2.7 times higher than that at 10 °C. At both temperatures, the GSA scaled consistently with body mass by an exponent of 0.802 (95% CI [0.759–0.846]), while RBC size scaled consistently with body mass by an exponent of 0.042 (95% CI [0.010–0.075]). The constant GSA scaling can explain the constant metabolic scaling as temperature increases, as metabolism may be constrained by fluxes across surfaces. The GSA at 10 °C was 1.2 times higher than that at 25 °C, which suggests that the constraints of GSA on the metabolism of black carp is induced by the higher temperature. The RBC size at 10 °C was 1.1 times higher than that at 25 °C. The smaller RBC size (a larger surface-to-volume ratio) at higher temperature suggests an enhanced oxygen supply and a reduced surface boundary limit on $b_R$, which offset the negative effect of temperature on $b_R$.

Subjects Aquaculture, Fisheries and Fish Science, Ecology, Zoology
Keywords Metabolism, Allometric, Fish, Gill, Red blood cell

## INTRODUCTION

The relationship between the metabolic rate (MR) and body mass (M) of organisms can be expressed as a power-law function, MR = $aM^b$, where $a$ is a constant and $b$ is the scaling exponent. The mechanisms of metabolic scaling have been attributed to the ontogenetic changes in the components of active organs (the organ size theory)

Corresponding author
Yiping Luo, luoguo@swu.edu.cn

(*Itazawa & Oikawa, 1983*; *Oikawa, Takemori & Itazawa, 1992*), the fractal patterns of resource distributions (the metabolic theory of ecology) (*West, Brown & Enquist, 1997*; *Brown et al., 2004*), the limitations on surfaces for mass and/or energy exchange (the surface area (SA) hypothesis) (*Rubner, 1883*; *Okie, 2013*), the ontogenetic change in the ratio of cell SA to volume (cell metabolic hypothesis) (*Davison, 1955*; *Kozłowski, Konarzewski & Gawelczyk, 2003*; *Starostová et al., 2009*), and the effects of SA vs. the volume-related metabolic processes of organisms mediated by metabolic level (the metabolic level boundaries hypothesis) (*Glazier, 2005*, *2008*, *2009*, *2010*). However, the underlying mechanisms of metabolic scaling remain unclear. Recently, a growing number of studies have implied that, in addition to the SA and metabolic level of whole organisms, SA at the organ level and the cellular level and their scaling may contribute to the allometric scaling of MR (*Okie, 2013*; *Hirst, Glazier & Atkinson, 2014*; *Luo et al., 2015*, *2020*; *Gillooly et al., 2016*; *Li et al., 2018*). It is of interest to investigate whether the changes in SA at the organ and cellular levels affect the metabolic scaling of the whole body of an organism.

Temperature is a major factor that affects MR and its scaling with body mass. Generally, the resting MR (RMR) of ectotherms increases with increasing temperature (*Jobling, 1994*; *Clarke & Johnston, 1999*). In fish, RMR increases with temperature by a coefficient ($Q_{10}$) ranging from 1.65 to 2.70 (*Jobling, 1994*; *Clarke & Johnston, 1999*; *Peck, Buckley & Bengtson, 2005*; *Ohlberger, Staaks & Hölker, 2007*). However, the changes in the exponent $b$ of the RMR ($b_R$) with temperature vary among species. In most cases, $b_R$ is reduced by increasing temperature (*Xie & Sun, 1990*; *Ohlberger, Staaks & Hölker, 2007*; *Ohlberger et al., 2008*, *2012*; *Killen, Atkinson & Glazier, 2010*; *Li et al., 2018*), which can be explained according to the metabolic level boundaries hypothesis (*Glazier, 2005*, *2008*, *2009*, *2010*, *2014a*, *2014b*). At high temperatures, high metabolic demand results in metabolic scaling being primarily determined by fluxes across surfaces (scaling as $M^{2/3}$). However, at low temperatures, low metabolic demand can be met by surface-related processes, and therefore metabolic scaling is determined more by the energy demand required to sustain the tissues (scaling as $M^1$). Thus, $b_R$ should decrease as temperature increases (*Glazier, 2005*, *2008*, *2009*, *2010*, *2014a*, *2014b*). However, increased and/or unchanged $b_R$ with temperature was observed in studies of the gastropod *Littorina littorea* (*Newell, 1973*), cuttlefish (*Sepia officinalis*) (*Melzner, Bock & Pörtner, 2007*; *Grigoriou & Richardson, 2009*), krill (*Euphausia pacifica*) (*Paranjape, 1967*), and several salamanders (*Gifford, Timothy & William, 2013*). In fishes, fluctuating and/or unchanged $b_R$ with temperature has been reported in several species of cyprinids and was speculated to be due to species-specific phenotypically plastic responses or to increasing spontaneous activity with temperature (*Hölker, 2003*, *2006*; *Luo & Wang, 2012*; *Ohlberger et al., 2012*). It remains unclear why $b_R$ can change with temperature in different patterns.

Previous studies have shown that the gill surface area (GSA) of fish may vary with temperature. For example, a thermally plastic increase in GSA has been observed in several species of cyprinids and has been suggested to be an adaptation of the fish to cope with its high metabolic demand with limited oxygen availability in warmer water (*Sollid, Weber & Nilsson, 2005*; *Sollid & Nilsson, 2006*; *Chen et al., 2019*). However, temperature may not

affect the mass scaling exponents of GSA ($b_G$) of fish (*Li et al., 2018*). It remains under debate whether GSA reflects metabolic demands (*Lefevre, McKenzie & Nilsson, 2017*, *2018*) or, in contrast, is a constraint on the metabolic rates of fish (*Pauly, 1981*; *Pauly & Cheung, 2018a*, *2018b*). In any case, GSA and its scaling should be relevant to metabolic rate and metabolic scaling. For example, $b_G$ is very close to $b_R$ in several species of carp, especially when controlling the effect of ventilation rate (*Luo et al., 2020*). Therefore, the determination of the GSA of fish at different temperatures may supply explanations for the effect of temperature on $b_R$.

The cell sizes of ectotherms can vary with temperature. The reduced cell size at warmer temperatures may increase the relative SA and improve the oxygen supply (*Audzijonyte et al., 2019*). The red blood cell (RBC) size is usually adopted as a proxy for the general cell size of an organism (*Starostová et al., 2009*, *2013*; *Luo et al., 2015*). Many studies have observed that the sizes of RBCs decrease with increasing temperature in ectotherms (*Van Voorhies, 1996*; *Goodman & Heah, 2010*; *Hermaniuk, Rybacki & Taylor, 2016*). It remains unclear whether the thermally reduced cell size compensates for the negative effect of temperature on $b_R$. The RBC size of fish generally scales with body mass by a very small exponent ($b_C$, 0–0.1) with a weak correlation with body mass (*Huang et al., 2013*; *Zhang et al., 2014*; *Luo et al., 2015*). It can be expected that temperature change may not induce a remarkable change in the $b_C$ of fish.

It can be predicted from the above backgrounds that the thermal plastic changes in SA at both the organ and cellular levels may change the limitation of the exchange surface on metabolic flows and therefore change the influence of temperature on metabolic scaling. If SA decreases with temperature, the limiting effect of SA further intensifies the negative effect of temperature on metabolic scaling, which results in decreased $b_R$ values as temperature increases. In contrast, if SA increases with temperature, the reduced limiting effect of SA compensates for the negative effect of temperature on metabolic scaling, which may result in unchanged and/or increased $b_R$ values as temperature increases.

Black carp (*Mylopharyngodon piceus*), a freshwater cyprinid fish native to rivers and lakes of China, was selected as the experimental animal in the present study. This species has been introduced into multiple regions of the world, including Europe, West-Central Asia and North America, and has been suggested to have serious adverse impacts on native aquatic communities due to predation on mollusc species (*Nico & Neilson, 2019*). This fish is tolerant of water temperatures from approximately 0.5–40 °C, and its optimum water temperatures range from 18 to 30 °C for adults (*CABI, 2019*). Information on the physiological response of black carp to temperature change would be useful for understanding its wide spread and for controlling its negative impacts on aquatic ecosystems. However, little is known regarding the effects of temperature on the physiology of this fish. A previous study showed that black carp could remodel their gill morphology to increase GSA under hypoxic conditions (*Dhillon et al., 2013*). Our recent studies have determined the metabolic scaling of black carp and suggested that both GSA and RBC size may affect the metabolic scaling of carp (*Lv et al., 2018*;

*Luo et al., 2020*). It remains unclear how temperature affects the RMR and $b_R$ of black carp and how temperature affects its GSA and RBC size and scaling.

The present study aimed to examine whether metabolic scaling in black carp changes with temperature and to determine the link between metabolic scaling and SA at the organ and cellular levels at different temperatures. Therefore, the RMR, GSA and RBC size of black carp were determined at 10 °C and 25 °C, and their scaling exponents were analyzed. We hypothesized that as temperature increases, (1) RMR increases, and $b_R$ decreases; (2) GSA increases, and $b_G$ does not change; and (3) RBC size decreases, but $b_C$ does not change.

## MATERIALS AND METHODS

### Experimental processes

Black carp (body mass ranging from 5 to 250 g, $n$ = 70) were obtained from Xiema aquaculture farm in Beibei, Chongqing, China and transferred to the laboratory. All experimental processes were performed according to the ethical requirements for animal care of the School of Life Sciences of Southwest University, China (LS-SWU/012/2016), and the requirements for the living environment and housing facilities for laboratory animals in China (Gb/T14925-2001). The fish were acclimated in a rearing system at their original temperature (17.0 °C) for 2 weeks, and the variations in temperature were kept within ±0.2 °C using a heater (JRB-250, Sunsun CO., Ltd., Zhejiang, China) regulated by an automatic temperature controller (PY-SM5, Pinyi CO., Ltd., Zhejiang, China). The dissolved oxygen concentration was kept above 95% saturation by continuous air pumping. Water temperature and oxygen concentration were checked twice daily (08:30 and 18:30) using a fiber-optic oxygen sensor probe connected with a portable multimeter (HQ30D, Hach Company, Loveland CO, USA). The ammonia concentration of the rearing water was monitored once daily (08:30) using the salicylate-hypochlorite method (*Verdouw, Van Echteld & Dekkers, 1978*). One-third of the volume of the rearing water was refreshed with aerated tap water every day (09:00) to keep the ammonia-N level below 0.01 mg L$^{-1}$. The photoperiod was 12h:12h light: dark. During acclimation, the fish were fed a commercial diet to satiation once daily (18:00), and the residual pellets were sucked out with a siphon one hour after feeding to ensure that the water was clean. The main chemical composition of the commercial diet was 12.5% moisture, 33.0% protein, 3.0% fat and 10.0% digestible carbohydrate (Tongwei Company, Sichuan, China). At the end of acclimation, the fish were randomly divided into two groups ($n$ = 35 for each group) for the temperature treatments. The fish of each group were reared in five tanks with seven individuals per tank. The tank sizes were 80 cm × 60 cm × 50 cm, and the water volume was 192 L in each tank. The water in each tank was recirculated through a filtration system (HBL803, Sunsun CO., Ltd., Zhejiang, China). One group was warmed to 25 °C by 1 °C per day using a heater (JRB-250, Sunsun CO., Ltd., Zhejiang, China) and an automatic temperature controller (PY-SM5, Pinyi CO., Ltd., Zhejiang, China). The other group was cooled by 1 °C per day to 10 °C using a cooling water machine (CW1000A, Risheng CO., Ltd., Guangdong, China) and an automatic temperature controller (PY-SM5, Pinyi CO., Ltd., Zhejiang, China).

Thereafter, the fish were reared at 10.0 ± 0.2 °C and 25.0 ± 0.2 °C for 2 weeks. The water quality monitoring and feeding processes were the same as those adopted during the abovementioned acclimation period. As one individual died during the 25 °C trial period, the final numbers of fish specimens were 34 for the 25 °C treatment and 35 for the 10 °C treatment.

The oxygen consumption rate was measured by a continuous flow-through respirometer and was used to represent the MR of individual fish (*Wang et al., 2012*). The structure of the respirometer is described in *Fu, Xie & Cao (2005)*. At the end of the temperature trial period, the fish were fasted for 48 h and individually placed into the respirometer chambers overnight before the determination of oxygen consumption. Chambers of different sizes (0.03, 0.13, 0.52, 0.86 and 1.20 L) were used depending on the size of the experimental individual, ensuring that the fish body could stretch to its full length but could not be inverted. Fifteen chambers were used at the same time for measurements, and one chamber without fish was used as the control to determine the background oxygen consumption. All chambers were placed in a water bath (110 cm × 70 cm × 30 cm). The visual field of the fish was shielded by the water bath to avoid visual disturbance by the experimental operation. The water flow rate was adjusted by regulating the valve at the inlet of each chamber prior to measurements. The outflow water from the respiration chamber was collected using a volumetric flask of a given volume, for which the time duration was recorded by a stopwatch. Then, the water flow velocity ($v$, l h$^{-1}$) was calculated. The flow rate was regulated to ensure that the dissolved oxygen in the outlet water of each breathing chamber was approximately 0.5–1 mg L$^{-1}$ lower than that of the control chamber but higher than 70% saturation concentration to avoid stress on the experimental fish due to lack of oxygen (*Blaikie & Kerr, 1996*; *Fu & Xie, 2004*). The dissolved oxygen concentration was measured at the outlet of the chamber using a probe connected with a portable multimeter (HQ30D, Hach Company, Loveland CO, USA). The oxygen consumption rate of each fish ($\dot{M}O_2$, mg O$_2$ h$^{-1}$) was calculated by the following formula:

$$\dot{M}O_2 = \Delta O_2 \times v$$

where $\Delta O_2$ is the difference between the dissolved oxygen concentrations (mg O$_2$ L$^{-1}$) of the outlets of the experimental chamber and the control chamber, and $v$ is the water flow rate through the chamber (L h$^{-1}$). The individual oxygen consumption rate was measured hourly from 09:00 to 15:00, and the mean value of the lowest two measurements was used as the RMR for that individual. The mean value of $Q_{10}$ of RMR over the range 10–25 °C was calculated by $Q_{10} = [(RMR_{25}/RMR_{10})]^{10/15}$, where $RMR_{25}$ and $RMR_{10}$ are the mass-corrected mean values of the RMRs at 25 °C and 10 °C, respectively.

After the determination of RMR, the fish were anesthetized for blood and gill arch sampling with 0.15 g L$^{-1}$ tricaine methanesulfonate (MS-222) for five minutes. Blood was taken from the caudal vessel using an anticoagulant-treated (0.04 mg mL$^{-1}$ sodium heparin) syringe and was transferred to an anticoagulant-treated centrifuge tube on ice. The blood sampling procedure was finished within 1 min. Then, blood smears were made

immediately. A blood smear failed for one individual at 25 °C because of the difficulty of sampling blood from a very small individual. The left four gill arches were dissected, kept in Bouin's fluid for 2 days and stored in 70% alcohol for several days (less than 1 week) before the GSA was measured. After blood and gill arch sampling, the fish were killed while anesthetized by quick freezing in liquid nitrogen since the fish were not capable of suffering while unconscious. Since the experimental animals were farmed fish, criteria for euthanizing the animals did not need to be established prior to the planned end of the experiment.

Blood smears were made with Wright–Giemsa dye solution (*Gao et al., 2007*; *Huang et al., 2013*). Fifty erythrocytes were randomly selected from the stained blood smears to determine the cell length (LC, μm) and cell width (WC, μm) using a microscope (EV5680B, Aigo Company, China) and Image-Pro Plus 6.0 software (Media Cybernetics, Rockville, MD, USA). The RBC area ($S_{RBC}$, μm$^2$) was calculated as $S_{RBC} = LC \times WC \times \pi/4$. The GSAs were determined according to the methods described by *Li et al. (2018)*. Briefly, the GSA was calculated by a standardized gill morphology assessment (*Hughes, 1966*, *1984*). The filament number on each gill arch was counted, and six filaments were sampled from each arch to measure the filament length and the secondary lamellae frequency. Three secondary lamellae were chosen from each of the six sampled filaments of the second arch to determine the bilateral areas of the lamella. The bilateral lamellae areas were determined using a microscope (EV5680B, Aigo Digital Technology Co., Beijing, China) and Image-Pro Plus 6.0 software (Media Cybernetics, Rockville, MD, USA). The total length of all the gill filaments (TL, mm), the mean lamella frequency (LF, mm$^{-1}$), and the mean lamella area (LA mm$^2$) were obtained. The GSA (mm$^2$) of an individual was calculated by the following formula: GSA = 4 × TL × LF × LA. The GSA determination failed for one individual at each temperature because of the difficulty of working with very small individuals. The raw data are in the Supplemental Files.

## Statistical analysis

The data were calculated using Microsoft Excel 2003 (Microsoft Corporation, Redmond, WA, USA) and were analyzed using R (*R Development Core Team, 2018*). The original data for variables including RMR, $S_{RBC}$ and GSA were log10 transformed. An ANCOVA (aov) was performed to analyze the effects of temperature and body mass on RMR, $S_{RBC}$, and GSA, using temperature as the categorical factor and body mass as the covariate. If the interaction between body mass and temperature was significant, ordinary least squares regression (lm) was performed on the RMR, $S_{RBC}$ and GSA data with body mass as the independent variable for each temperature. If the interaction between body mass and temperature was not significant, the regression analysis was performed on the RMR, $S_{RBC}$ and GSA data with both body mass and temperature as independent variables. The linear regression slopes for the RMR and the GSA were compared at each temperature. The *b* values were expressed with 95% confidence intervals (CIs) and compared to given values using a *t*-test. The ordinary least squares regression (lm) was used to analyze the mass-independent correlations between the residual values of RMR, $S_{RBC}$ and GSA. The significance level was $P < 0.05$.

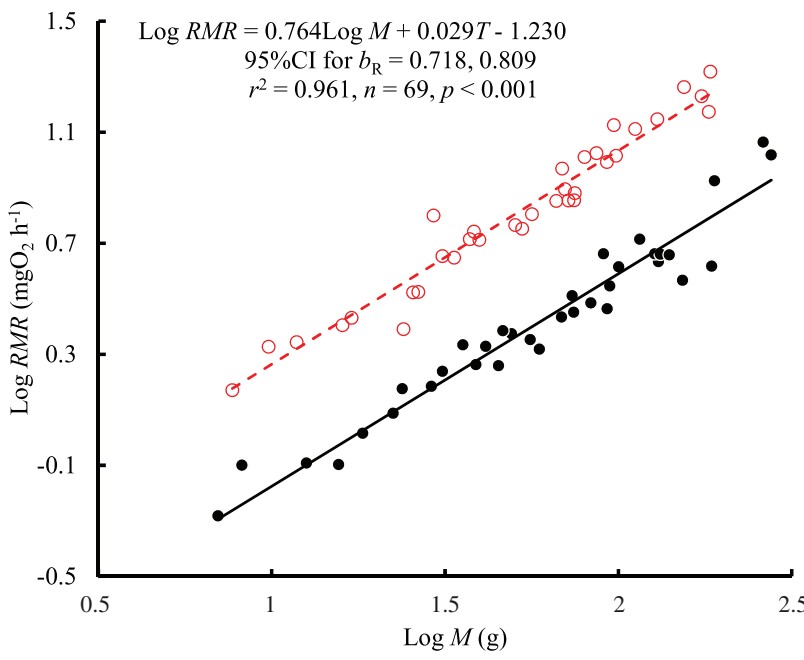

**Figure 1  The relationship between resting metabolic rate (RMR, mg O$_2$ h$^{-1}$) and body mass ($M$, g) of the black carp *Mylopharyngodon piceus* treated by two temperatures ($T$, °C) for 2 weeks.** The original data for variables including RMR and $M$ were log10 transformed. No significant interaction between $M$ and $T$ on RMR was observed using the general linear model (glm). Then, the ordinary least squares regression (lm) was performed on the RMR with $M$ and $T$. Red open circles and dashed line: 25 °C; black filled circles and solid line: 10 °C.                                

## RESULTS

The results showed that both body mass ($F$ = 1,021.04, $P$ < 0.001) and temperature ($F$ = 648.44, $P$ < 0.001) affected the RMR of black carp, with no significant interaction between body mass and temperature ($F$ = 1.02, $P$ = 0.317). Therefore, a consistent $b_R$ (0.764, 95% CI [0.718–0.809]) between temperatures was obtained based on the regression with log body mass and temperature as independent variables (Fig. 1). The value of $b_R$ was significantly larger than 2/3 ($t$ = 4.31, $P$ < 0.001) but not larger than 3/4 ($t$ = 0.62, $P$ = 0.537). The mass-specific RMR ranged from 81.8 to 216.6 mg O$_2$ kg$^{-1}$ h$^{-1}$ at 25 °C and from 22.3 to 96.9 mg O$_2$ kg$^{-1}$ h$^{-1}$ at 10 °C. The RMR at 25 °C was 2.7 times higher than that at 10 °C when corrected to the mean values for the body masses, representing a Q$_{10}$ value of 1.95.

The interaction between body mass and temperature had no significant effect on GSA ($F$ = 0.005, $P$ = 0.943). Both body mass ($F$ = 1389.42, $P$ < 0.001) and temperature ($F$ = 33.27, $P$ < 0.001) significantly affected GSA. GSA scaled consistently with body mass at both temperatures (Fig. 2), and $b_{GSA}$ (0.802, 95% CI [0.759–0.846]) was significantly higher than 2/3 ($t$ = 6.21, $P$ < 0.001) but was not different from the $b_R$ value ($F$ = 1.87, $P$ = 0.176 at 10 °C; $F$ = 1.39, $P$ = 0.24 at 25 °C). When controlling for the effects of body mass, the GSA at 10 °C was 1.2 times higher than that at 25 °C. No significant correlation was found between GSA and RMR when controlling for body mass ($r^2$ = −0.0134, $P$ = 0.719) (Fig. S1).

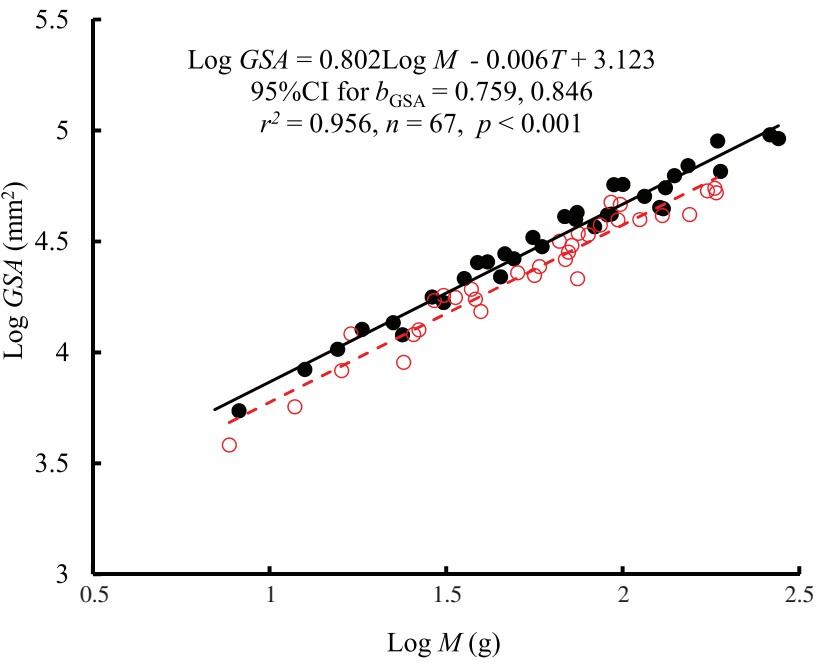

$$\text{Log } GSA = 0.802 \text{Log } M - 0.006T + 3.123$$
$$95\%\text{CI for } b_{\text{GSA}} = 0.759, 0.846$$
$$r^2 = 0.956, \ n = 67, \ p < 0.001$$

**Figure 2 The relationship between gill surface area (GSA, mm$^2$) and body mass ($M$) of the black carp** *Mylopharyngodon piceus* **treated by two temperatures ($T$, °C) for 2 weeks.** No significant interaction between $M$ and $T$ on GSA was observed using the general linear model (glm). Then, the ordinary least squares regression (lm) was performed on the GSA with $M$ and $T$. Red open circles and dashed line: 25 °C; black filled circles and solid line: 10 °C.

The interaction between body mass and temperature had no significant effect on $S_{\text{RBC}}$ ($F = 0.01$, $P = 0.921$). Both body mass ($F = 7.68$, $P = 0.007$) and temperature ($F = 12.61$, $P < 0.001$) significantly affected $S_{\text{RBC}}$. $S_{\text{RBC}}$ scaled consistently with body mass at both temperatures by a $b_{\text{C}}$ of 0.042 (95% CI [0.010–0.075]) (Fig. 3). However, there was only a weak correlation between $S_{\text{RBC}}$ and body mass ($r^2 = 0.217$). When controlling for the effects of body mass, the $S_{\text{RBC}}$ at 10 °C was 1.1 times higher than that at 25 °C. No significant correlation was found between $S_{\text{RBC}}$ and RMR when controlling for body mass ($r^2 = -0.0133$, $P = 0.728$) (Fig. S2).

## DISCUSSION

In the present study, the RMR of black carp increased with increasing temperature. The $Q_{10}$ value (1.95) was within the $Q_{10}$ value range (1.65–2.70) of the RMR of most previously reported fish (*Jobling, 1994*; *Peck, Buckley & Bengtson, 2005*; *Ohlberger, Staaks & Hölker, 2007*), suggesting that the metabolic level of black carp is moderately sensitive to temperature. However, inconsistent with our prediction, the results showed that the effect of body mass on the RMR of black carp was independent of temperature, that is, the increased metabolic level with increasing temperature did not reduce the $b_{\text{R}}$ of the black carp.

In addition, in contrast to our prediction, the GSA of black carp decreased with increasing temperature. Therefore, black carp does not increase its GSA to cope with

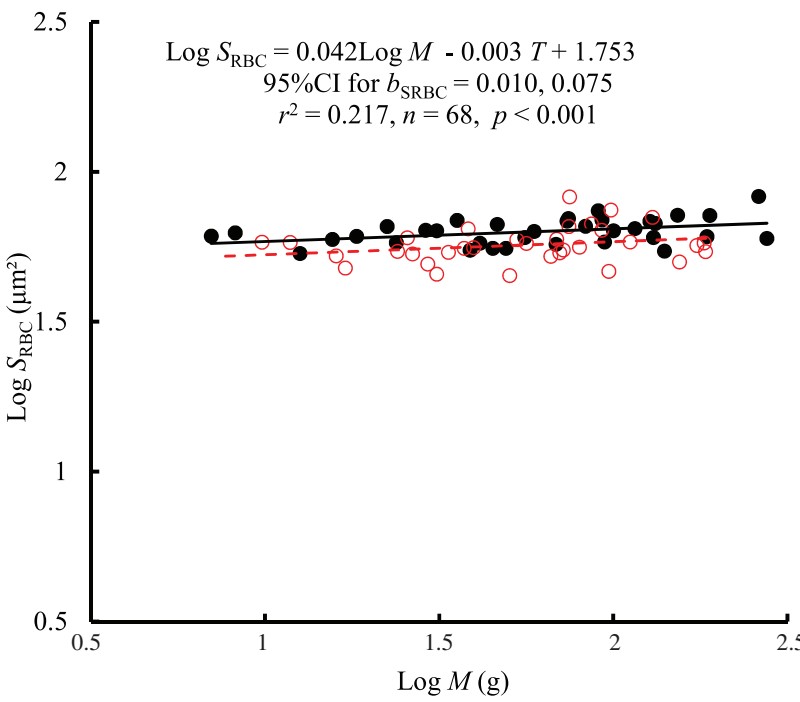

**Figure 3 The relationship between red blood cell size ($S_{RBC}$, μm²) and mass of the black carp _Mylopharyngodon piceus_ treated by two temperatures (_T_, °C) for 2 weeks.** No significant interaction between _M_ and _T_ on $S_{RBC}$ was observed using the general linear model (glm). Then, the ordinary least squares regression (lm) was performed on the $S_{RBC}$ with _M_ and _T_. Red open circles and dashed line: 25 °C; black filled circles and solid line: 10 °C.

the high metabolic demand in warmer water, in contrast to previously reported fish (_Sollid, Weber & Nilsson, 2005_; _Sollid & Nilsson, 2006_; _Chen et al., 2019_). Changes in the GSA of fish have relevance not only to oxygen supply/demand but also to ion exchange and parasitic infections (Pauly, 1981; Lefevre et al., 2017, 2018; Pauly & Cheung, 2018a, 2018b). Reducing GSA can help decrease losses of ions and risk of infections in warmer water. In addition to GSA, the ventilation rate could regulate oxygen supplies (_Luo et al., 2020_; _Xiong et al., 2020_). Therefore, an increase in RMR with increasing temperature is not necessarily accompanied by an increase in GSA. The results suggest that GSA may not reflect metabolic demands, as proposed by Lefevre et al. (2017, 2018). In contrast, the constraints of GSA on the metabolism of black carp may be enhanced at higher temperatures. However, the $b_R$ of the fish did not decrease with increasing temperature. There are several possible causes that can counteract the negative effect of temperature on the $b_R$. First, the spontaneous activity of the fish may increase at high temperatures, which causes the metabolic rate to be increasingly dominated by the energy demands of muscular tissue, which is directly proportional to muscle mass (_b_ = 1) and can result in a positive effect on $b_R$ at high temperatures (_Luo & Wang, 2012_; _Glazier, 2014b_). Second, similar to previous results in many ectotherms (_Van Voorhies, 1996_; _Goodman & Heah, 2010_; _Hermaniuk, Rybacki & Taylor, 2016_), our study found that black carp have a smaller RBC size (a larger surface-to-volume ratio) at higher temperatures, which may enhance the oxygen supply and weaken the effect of the surface boundary

limit on $b_R$ (*Glazier, 2005, 2010; Luo et al., 2015*). Thus, the negative effect of temperature on $b_R$ was compensated. Third, the metabolic rate should be determined by fluxes across surfaces; therefore, metabolic scaling should be determined by SA scaling (*Wells & Pinder, 1996a, 1996b; Reich, 2001; Yoshiyama & Klausmeier, 2008; Kooijman, 2009; Phillips et al., 2009; Glazier, Hirst & Atkinson, 2015; Luo et al., 2020*). Therefore, when SA scaling remains unchanged as temperature increases, metabolic scaling should remain constant. Accordingly, our results showed no difference between the $b_R$ and $b_{GSA}$ of black carp at either temperature. Indeed, even though the $b_R$ of many fishes decrease with increasing temperature (*Xie & Sun, 1990; Ohlberger, Staaks & Hölker, 2007; Ohlberger et al., 2008, 2012; Li et al., 2018*), a fluctuating and/or unchanged $b_R$ was observed in several fishes, for example, *Coreius guichenoti* (*Luo & Wang, 2012*), *Abramis brama* and *Rutilus rutilus* (*Hölker, 2003, 2006; Ohlberger et al., 2012*).

The temperature-induced change in exchange surfaces at the individual level may play a role in the metabolic scaling of fish (*Li et al., 2018*). In the present results, the different GSA and $S_{RBC}$ at different temperatures suggest that plasticity in exchange surfaces may occur both at the individual level and at the cellular level. The cell metabolic hypothesis proposed that metabolic level negatively correlates with the cell size of organisms (*Davison, 1955; Kozłowski, Konarzewski & Gawelczyk, 2003; Starostová et al., 2013*). However, the present results showed no negative correlation between RMR and $S_{RBC}$ in black carp. The results were not surprising, as correlations between RMR and $S_{RBC}$ vary among fish species (*Maciak et al., 2011; Huang et al., 2013; Zhang et al., 2014; Luo et al., 2015; Lv et al., 2018*). According to the cell metabolic hypothesis, $b_R$ equals 1 when the cell size scales as $M^0$ and 2/3 when the cell size scales as $M^{2/3}$ (*Davison, 1955; Szarski, 1983; Kozłowski, Konarzewski & Gawelczyk, 2003*). In the present study, the RBC size scaled with body mass by a very small exponent (0.042), which predicts a metabolic scaling exponent very close to 1 according to the cell metabolic hypothesis. However, the observed $b_R$ value in black carp was 0.764. Therefore, ontogenetic changes in cell size may not contribute much to the metabolic scaling of black carp.

## CONCLUSION

In conclusion, the present study provides comparative data on SA parameters and RMR at the organismal and cellular level, as well as the mass scaling exponents of black carp at different temperatures. These RMRs increased while both GSA and $S_{RBC}$ decreased with increasing temperature. No significant difference in $b_R$, $b_G$, or $b_C$ exists between temperatures. This indicates that the negative effect of temperature on the metabolic scaling exponent is offset by other factors, for example, increasing spontaneous activity, decreasing cell size and constant GSA scaling at higher temperatures.

## ACKNOWLEDGEMENTS

We thank Mr. Bo Zhang for his help with fish collection. We sincerely thank the anonymous reviewers for their helpful suggestions and comments.

### Funding

This work was supported by the National Natural Science Foundation of China (No. 31672287). The funders had no role in study design, data collection and analysis, decision to publish, or preparation of the manuscript.

### Grant Disclosures

The following grant information was disclosed by the authors:
National Natural Science Foundation of China: 31672287.

### Competing Interests

The authors declare that they have no competing interests.

### Author Contributions

- Qian Li performed the experiments, analyzed the data, prepared figures and/or tables, authored or reviewed drafts of the paper, and approved the final draft.
- Xiaoling Zhu performed the experiments, analyzed the data, authored or reviewed drafts of the paper, and approved the final draft.
- Wei Xiong analyzed the data, authored or reviewed drafts of the paper, and approved the final draft.
- Yanqiu Zhu analyzed the data, authored or reviewed drafts of the paper, and approved the final draft.
- Jianghui Zhang analyzed the data, authored or reviewed drafts of the paper, and approved the final draft.
- Pathe Karim Djiba analyzed the data, authored or reviewed drafts of the paper, and approved the final draft.
- Xiao Lv conceived and designed the experiments, performed the experiments, prepared figures and/or tables, and approved the final draft.
- Yiping Luo conceived and designed the experiments, analyzed the data, prepared figures and/or tables, authored or reviewed drafts of the paper, and approved the final draft.

### Animal Ethics

The following information was supplied relating to ethical approvals (i.e., approving body and any reference numbers):

The School of Life Sciences, Southwest University provided full approval for this research (LS-SWU/012/2016).

### Data Availability

The raw data are available in the Supplemental Files.

## Supplemental Information

Supplemental information for this article can be found online at http://dx.doi.org/10.7717/peerj.9242#supplemental-information.

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
