# Peer review of "Effects of temperature on metabolic scaling in black carp"

_PeerJ, doi:10.7717/peerj.9242_

## Round 0.1 · original submission · Major Revisions

Your manuscript has to be deeply improved. More concretely, your experimental design is not clear in your manuscript, and you need to discuss the drawbacks or limitations of your experimental design frankly in your Discussion.

·

Basic reporting

No comments

Experimental design

No comments

Validity of the findings

No comments

Additional comments

GERENAL COMMENTS

The manuscript entitled “Effects of temperature on metabolic scaling in black carp” looked at examine whether metabolic scaling in black carp changes with temperature and to identify the link between metabolic scaling and surface area at the organ and cellular levels at two temperatures (10 vs 25 oC). Authors have found that resting metabolic rate, gill surface area, and red blood cell size varied with temperature, but the temperature’s effect was opposite between resting metabolic rate and gill surface area, and red blood cell size. However, all the parameters scaled consistently with body mass across temperature. This is a well written manuscript that was easy to follow and, although limited in its scope, was conclusive in its findings, and has interesting implications. However, this manuscript needs to be clarified in main sections (e.g., materials and methods) that I concerned as follows.

SPECIFIC COMMENTS

Line 27, change ‘095%’ to ‘95%’.

Lines 53-59, it is too long due to their un-consistent meanings. I suggest authors cut into two sentences.

Lines 65-66, it seems not to be appropriate presented at the end of the second paragraph, or it needs to be rewrote.

Line 80, delete ’significantly’.

Line 98, please provide main content for diet if available.

Lines 103-104, it is not clear whether fish from each temperature treatment were all kept within one tank. Fish should be kept in at least two rearing tanks to avoid pseudoreplication.

Line 106, it is too short for fish temperature acclimation, but would be better at least four weeks.

Line 111, is it sufficient using 48 h for 10 degree? Cite reference if available. Fish would extend their digestion and absorption (i.e., SDA) period at low temperature.

Line 112, what’s size for these chamber used for fish of different size? Please clarify here.

Lines 122-124 and elsewhere: a dot above the ’M’ in MO2 is missing in most places.

Line 170, check F-value again since it is so large.

All figures are not vectors so authors should provide more clearer figures if available.

·

Basic reporting

The manuscript reports on temperature effects on metabolic scaling, gill surface area (GSA) scaling and the scaling of red blood cells (RBC) in black carp (Mylopharyngodon piceus). The authors found that at high temperatures resting metabolic rate (RMR) was elevated, while both GSA and RBC size was reduced compared to the low temperature treatment. Li et al show a consistent scaling exponent with fish body mass of all tested parameters, independent of temperature. Generally, I think that the findings of the study are interesting, and the hypotheses are scientifically relevant. However, significant changes and additions need to be made in order to meet the publication standard of PeerJ.
1 BASIC REPORTING
Firstly, I think the abstract would benefit from a more extensive discussion of the presented results (currently only one sentence L29-31). An explanation on how RBC size and gill surface area offset the negative effect of temperature on the metabolic scaling exponent would be interesting.
In some parts I find the reporting language slightly confusing, as some sentences are not well structured and the terminology is not consistently used throughout the manuscript. The whole manuscript language needs to be revised. Some examples below:
L53-59: the structure of this sentence is hard to follow and hence hard to understand. I suggest breaking it up into multiple sentences and adding some information.
L73-76: repeated word for word from abstract (line 15-18). Consider rephrasing.
L80-82: It is not clear what the authors are trying to say. This sentence needs rewording
L157-159: be consistent with the acronyms (SRBC vs SRBC) throughout the text.
L208-210: This sentence is hard to follow. I suggest breaking it up in two and provide more information.
L218-222: A very nested sentence, which doesn’t flow well. A lot of information is presented, but is hard to grasp in its current form. Phrasings like “surface area of the rate limiting metabolic structure (L219)” make it hard for the reader to follow the thoughts of the authors. This needs rephrasing.
L236-239: Again, a nested sentence which would profit from being broken up into two.
The introduction gives a good insight in the topic and prepares the reader for the data presented. It could be improved by a thorough language revision as mentioned above. I think the introduction would be advanced further by more background information on the species used (especially of the habitat and temperature window it occurs naturally). The end of the introduction needs to contain the hypotheses on the effect of temperature on the GSA and RBC size scaling, not only that of metabolic rate (L87-88). I also think the authors need to elaborate more on the importance of how the metabolic rate, RBC size and GSA of black carb scales with temperature.
Furthermore, neither in the introduction, nor in the discussion do Li at al mention the literature on the scaling of body mass and GSA with temperature from Pauly et al and the debates it caused (e.g. Lefevre, McKenzie and Nilsson (2017); Pauly (1981); Pauly and Cheung (2018)). I think the manuscript would greatly benefit from including these publications and from discussing the results with this background. Also the sentence in L205-208 needs a reference to support it, but besides that the literature presented in this manuscript is well referenced and relevant.
The structure of the article is well chosen and conforms to PeerJ standards. The conclusion however, needs to be more comprehensive than it currently is, more on that below. Also the term Q10 (L196) is firstly mentioned in the discussion, but had not been explained in the introduction. Furthermore the manuscript lacks the method of calculation of Q10 (should be in the methods section) and the presentation of the results (should be in the result section). Please amend.
The figures presented in this study are mostly relevant. Figure 3 and 5 are not essential for the narrative of the manuscript and hence could be moved to the supplementary material, in case space is an issue. Furthermore the figure captions need to be more descriptive (e.g. include scientific name of species, treatment, incubation time, statistical methods and so on). The figures need be understandable without information from manuscript, which currently they are not. Also Figure 4 contains only open circles, but the caption states there are filled ones. Please amend the figures accordingly.
I commend the authors for supplying a set of neatly organised raw data.

Experimental design

As of my knowledge the research presented in this study is original primary research, well within the scope of PeerJ. As I mentioned above, I think that the hypotheses need to be extended to also include the scaling of GSA and RBC. The knowledge gap the authors attempt to fill is well defined, however the relevance to justify the research needs to be stated more clearly.
The methods of this manuscript need extensive work in order to be considered for publication.
Firstly, the origin of the fish is unclear. Where have they been caught? Or are they reared in aquaculture? The authors mention that they were fished (L92) and later on Li et al say that their specimens are farmed (L136). Please specify.
The rearing facilities are not well described. This however is crucial for the study in order to be replicable. A detailed description of holding tanks (among others the volume, flow-through or recirculating system, filtration system, and most important how many replicate tanks per treatment) needs to be added, so reviewers are able to assess the experimental design. If the temperature treatments have not been replicated, than this would be a faulty experimental design, and would require admitting the design error in the text and discussing the potential issues with that. Furthermore, the rearing density of each replicate tank needs to be mentioned, as well as the supplier of the commercial feed. Also, it is unclear from the manuscript if temperature, ammonia and oxygen levels were measured during the acclimation and the trial period. Please include measuring techniques, equipment, and manufacturers. Further it is ambiguous how the temperature treatments were set, i.e. was the water heated/cooled? If yes mention the equipment used and its set up. In general I find 2 weeks of acclimatisation a very short period of time. A longer acclimatisation time might have led to different results. In L110 you use the word “acclimation period” for the time the fish were at their respective temperatures. Earlier on (L96) you say the fish were “acclimated in the rearing system at their original temperature”. This is confusing and I would suggest using the word “experimental or trial period” for the former instead, in order to clearly distinguish between the different experimental stages.
The set-up of the respirometers is also ambiguous. The authors need to specify the volume of the respiration chambers, and to which flow rate and how they adjusted the water inflow. Further measuring the water flow rate using a beaker seems to be a rather rough estimate to me. I suggest using more precise methods in the future. And did those measurements disturb the fish? How much were the fish moving in the respirometers and how much did that affect the readings when the authors only got one measure per hour? Please see Fig. 2 in (Clark, Sandblom & Jutfelt 2013) for further information on the potential issues of measuring oxygen consumption infrequently and discuss your results accordingly. Did the beaker oxygenate the water during the measurements? That would reduce apparent oxygen consumption in the most heavily consuming fish (lower oxygen levels give higher diffusion gradient), and perhaps that was manly the warm group?
The sampling method is well described. The duration from when the fish were anesthetised until blood sampling is missing though. Studies (Iwama, McGeer & Pawluk 1989; Phuong et al. 2017) show that fish stressed by anaesthetics can show a swelling of red blood cells/ higher haematocrit, which would lead to an erroneous measurement of RBC size. Was this considered in the design of the study? Also what was the time frame between sampling the blood and preparing the blood smears? And what substance did you use as anticoagulant? Please specify.
The sentence L133-134 is slightly confusing. It sounds like the authors transferred the gills to 70% alcohol (ethanol) right before measuring the GSA. Please verify if that was the method used, or if you stored the samples in ethanol until the analysis (e.g. for several days). Further, it sounds like the authors took three lamellae from every filament on arch two. This would be a very tedious analysis. Please give a more detailed description on which part of the organ you analysed. It is also unclear how you measured the bilateral lamellae area. Please provide details.

Validity of the findings

Most of the underlying data are provided by the authors (apart from data on Q10, see above), and hence the study is easily comprehensible. With the information given at hand, I am not able to comment on the statistical soundness of the present study. The missing information about the experimental design, especially concerning replicate tanks make a judgement of the scientific merit problematic. Generally, I find that the discussion has a very heavy focus on the scaling exponents, and therefore dismisses interesting results, such as the overall smaller GSA at high temperatures compared to the high RMR. I suggest to include a discussion of this point as well. The authors talk about how an increased growth rate can mask the negative effect of temperature on the scaling exponent of the RMR. If that would indeed be the case, upon termination of the experiment I would expect the high temperature group to have a larger mass than the low temperature group. Was this the case? If not, this might not be a valid explanation.
The data for the RBC size seem to have a very high spread, and therefore a low r2 of 0.221, which might indicate that there is only a weak correlation of RBC with body mass. This however is not mentioned in the manuscript. Furthermore, data of some specimens are missing in the raw data (GSA of LQY33 and RBC of HQY31), as well as GSA data from fish HQY34 does appear in the raw data, but does not seem to be in the Fig. 2. Please explain.
Finally, the conclusion should combine all findings, and not only those of the scaling exponent of the RMR. Please add a short paragraph about GSA and RBC.

---

## Round 0.2 · Minor Revisions

Thank you for improving your manuscript which still needs some minor modifications indicated in this mail.

·

Basic reporting

This revised MS is substantially improved, and the authors have addressed nearly all of the two reviewers' comments, as indicated by sufficient backgroud, umambiguous texts, more clear result descriptions and discussion presented within the MS.

Experimental design

no comment

Validity of the findings

no comment

·

Basic reporting

no comments

Experimental design

no comments

Validity of the findings

no comments

Additional comments

The manuscript entitled "Effects of temperature on metabolic scaling in black carp" reports on temperature effects on the scaling exponents of resting metabolic rate, gill surface area, and red blood cell size.
The quality of the manuscript improved significantly throughout the review process and I believe, after minor amendments listed below, it will be well suited for publication within the journal PeerJ.

Specific comments:
L28-29 This sentence is unclear: do you mean GSA?
L57 I suggest writing exponent b instead of b value
L141 I suggest writing “a commercial diet”, instead of “commercial diets”
L142 Please write “residual pellets” or “residual feed”, not “residual diets”
L166 Please give a short description of the setup (how big the water bath was (e.g. dimensions), and if all chambers sit in the same bath, or if several water baths were used).
L192 Saying the “first four gill arches” implies that black carp has more than four gill arches on each side. Is that true? If not, please refer to the left four gill arches.

---

## Round 0.3 · accepted · Accept

Dear Authors,

Thank you very much for improving your manuscript. I am pleased to confirm that your paper has been accepted for publication in PeerJ.

Thank you for submitting your work to this journal.